# Analysis of Ankle sEMG on Both Stable and Unstable Surfaces for Elderly and Young Women—A Pilot Study

**DOI:** 10.3390/ijerph16091544

**Published:** 2019-05-01

**Authors:** Monika Błaszczyszyn, Mariusz Konieczny, Paweł Pakosz

**Affiliations:** Physical Education and Physiotherapy Department, Opole University of Technology, 45-758 Opole, Poland; m.blaszczyszyn@po.opole.pl (M.B.); p.pakosz@po.opole.pl (P.P.)

**Keywords:** postural balance, surface electromyography, ankle joints

## Abstract

Body aging is frequently accompanied with numerous consequences such as increased tendency to falls, which may be connected not only with the impaired ability of keeping balance, but also with some limitations of the system responsible for the maintenance of balance. The aim of this study was defining the importance of muscle activity in the ankle joint when standing on stable and unstable surfaces with eyes open and closed as well as examining indications of the influence of body aging on the activity of the selected muscles. The study was carried out on a group of 20 healthy women divided into two age groups (aged 24 ± 3.6 years and aged 64 ± 4.2 years). The activity of muscles stabilizing the ankle joint while standing on different types of surfaces was tested with an EMG (surface electromyography) on both elderly and young women during trials with open and closed eyes. The tests showed higher activation in the evaluated muscles of the anterior group (tibialis anterior (TA), peroneus longus (PL)) of elderly women. For the posterior shin muscles of both groups, a higher activation was observed in the gastrocnemius lateralis muscle (GL) of elderly women, whereas gastrocnemius medialis muscle (GM) showed highly comparable activity. The results obtained in this work confirm the importance of proprioception training and muscle strength in the rehabilitation of older people as well as in prophylactic exercise aimed toward the prevention of ankle injuries.

## 1. Introduction

The ability to prevent sudden loss of balance depends on the ability to create appropriate mechanisms acting on the ground. This is of particular importance in populations of older people who are at risk of falls due to involutional changes in the human body [1]. Injuries related with falls of elderly people are a major cause of suffering and/or costs, especially as they affect 33% of adults aged 65 or older [2]. Also in accordance with some statistical data, half of elderly people fall repeatedly, and on average every tenth fall results in a serious head injury [3]. Such injuries may also frequently lead to hospitalization or even death. They are also responsible for the deterioration of quality of life and significant loss of independence [4].

The ability to prevent sudden loss of balance depends on being able to create appropriate mechanisms, as influenced by factors such as surface and relations between its size and stiffness. Static and dynamic balance is maintained by the vestibular system, the visual information system, and proprioception coming from sensory receptors located within muscles, joints, and tendons [5,6].

The most important limitation of balance in terms of the biomechanical aspects is the size and quality of the foot support base. All limitations connected with muscle strength, the scope of joint mobility, pain, and control of foot position influence the ability to keep balance [7] and, in addition, the limit of stability and individually taken strategies against loss of balance. One of the main movement strategies listed by researchers that seems to be really important when attempting to keep a balanced position on a hard surface is the ankle strategy [8]. Therefore, one of the mechanisms responsible for balance in the anterior-posterior direction is the muscle control of ankle stabilizers. It can be assumed that the reaction times of these muscles play a key role in controlling body balance in the standing position [9].

With the elongation of life, more and more involutionary changes are created, including increasing vulnerability of falling among older people during activities of daily living. It is an important matter that cannot be ignored by members of the elderly population, their families, and health services. Numerous studies have shown that one in three people older than 65 experiences on average at least one fall in a year [10]. The probability of injury as a result of a fall is 18% in women and 24% in men. Injuries from falls are most often limb injuries [11]. About 30% of elderly people who experienced a fall suffer from serious injuries, such as a hip fracture or head injury [12]. In addition, falls can cause injuries and traumatic experiences that can lead to limited activity, fear of the next fall, disability, and increased dependence on other people. The listed factors can decisively lead to decreased social and physical activity among the elderly [12,13]. It should also be highlighted that the presence of chronic diseases, such as diabetes and cardio-vascular diseases, is a factor that can increase the risk of people falling down [14]. Even in cases in which falls do not result in serious body injuries, the falls may often cause long-term states of anxiety that influence the stability and body posture as well as further life activity of the elderly people. The indicated factors limit the physical activity of these people, which, as a consequence, can lead to increased disability. In response to minor disturbing factors, balance can be restored by coordinated activity of muscles that stabilize the ankle in conditions that do not require an individual to raise his or her foot from the surface [15]. 

sEMG (surface electromyography), in connection with other biomechanic methods, indicates important information about muscle activity during performance of different types of activity with changeable load, angle positioning in joints, and changeable speed [9]. 

The aim of the present study was to define the importance of muscle activity in the ankle joint (the tibialis anterior (TA), peroneus longus (PL), gastrocnemius medialis (GM), gastrocnemius lateralis (GL)) of individuals who were standing on stable and unstable surfaces with eyes open (EO) and eyes closed (EC) as well as to examine the influence of body aging on the activity of the selected muscles. The authors hypothesized that the activity of ankle muscles can affect the ability to maintain a stable posture in static conditions on different types of surfaces.

## 2. Materials and Methods 

The study was conducted on a group of 20 healthy women, including 10 women aged 24 ± 3.6 years, with BMI (body mass index) 20.93 ± 1.8 and 10 aged 64 ± 4.2 years, with BMI 25.98 ± 2.6. The study was performed on the dominant limb. All participants declared the right foot as the dominant foot. The criteria for participation in the study were lack of: lower limb injuries, neuro-muscular diseases, orthopedic diseases, and fall history as well as lack of sight or hearing disorders and no existing problems with keeping balance. The participants were people who neither trained nor regularly did physical exercises.

The study consisted of six attempts. The duration of each attempt was 20 seconds on all types of surfaces with eyes open (EO) and eyes closed (EC), with 5-second breaks for the command “eyes closed”. The participants, one by one, performed an attempt standing on the hard stable surface with EO and EC, and then stood on a piece of foam with dimensions of 50 × 50 cm, height 20 cm, and density 40 kg/m^3^ with EO and EC. The last activity required participants to stand on a “seesaw” in a position allowed for the possibility of moving forwards and backwards with EO and EC (Figure 1). The participants received a command to keep a stable posture and look straight ahead, and were instructed not to raise their feet from the ground. 

The scope and goal of this study were approved by the Bioethics Committee of the Chamber of Physicians (Resolution No. 237 of 13 December 2016), and the study was performed in accordance with the guidelines defined in the Helsinki Declaration for the conduct of clinical trials in humans.

System EMG produced by Noraxon (Scottsdale, AZ, USA), which registers muscle activity, was used as a research tool; it is a so-called dynamic EMG for use in training conditions with wire communication between pre-amplifiers and the unit collecting signal. Digital signals registering EMG parameters were sent via telemetry to a computer. Prior to the procedure, the measurement spots on each participant’s body were prepared by removing hair so as to improve the electrical contact between the electrodes and the skin. The surface electrodes (Ag/AgCl) were situated on the belly of muscles between the motor points and the tendon origins along the midline of the muscles. After the skin preparation procedure, we marked electrode positions while participants were standing upright according to the guidelines of the European recommendations for SEMG (SENIAM) (see Figure 1). We used disposable electrodes Bio-Lead-Lok B R-LFO-300 (Bio-Lead-Lok B, Józefów, Poland) with a round pickup area; the distance between electrodes was 2.5 cm. The data analysis was performed with MyoResearch XP MT 400 programme ( Noraxo, Scottsdale, AZ, USA). The sampling rate was 1000 Hz. The root mean square (RMS) values of EMG signals were calculated for consecutive segments of 50 ms. Video material was registered with a camera synchronized with EMG record, which recorded 60 frames per second (MyoResearch XP MT 400). The coupling of EMG registration with video record was used by the authors to eliminate erroneous attempts.

Statistical analysis methods of Statistica 13 programme were applied in order to verify the accepted hypothesis. The group structure was characterized according to the arithmetical mean and one-standard deviation, whereas ANOVA with repeatable measurements was applied in order to define the differences between variables. The indices of the sEMG signals were subjected to the Shapiro–Wilk normality test. The distributions of the analyzed variables did not deviate from normality, thus dependent variables were split into two groups (young and elderly) × three surfaces (hard, foam, seesaw) × two conditions (with eyes open and eyes closed) and analyzed via ANOVA (Statistica 13.0, Publisher, City, State abbrev if USA, Country) with repeated measures of the last two factors. Selected pairwise comparisons were explored using follow-up analyses (Tukey test). The level of significance was set at *p* < 0.05.

## 3. Results

The application of the analysis of the experiments with repeatable measurements allowed for the reactions of muscle activity (ratio of synergistic and antagonistic muscles of the ankle joint Figure 1, Figure 2, Figure 3 and Figure 4) to be shown among the same persons in repeated measurements on surfaces with differing levels of stability, and also allowed for the introduction of an additional factor in the form of the change in visual perception (EO and EC). The EMG relation is presented in percentage values in the diagrams in this paper. The first aspect of the discussed factors is the differences in EMG activation on the unstable surface under the conditions of normal and limited visual perception. 

Detailed statistical data of the main effects and the interaction between the effects (tested EMG) are presented in Table 1.

The research indicates that the EMG ratios between young and elderly people for EO versus EC attempts were not statistically important. However, the lack of importance is not alarming as, in terms of the effect of age, the differentiation in relation to surface stability was not introduced. An important issue is the relationship involving EMG activation. Older people generated almost 80% in the sample with EO, which was at the same level as in the case of the EC attempt. In young people, the ratio was 65% (Figure 2).

The differences turned out to be statistically important (*p* = 0.001). In stable conditions (rest), EMG activation with EO was in 89% of cases similar to the activity of the examined muscles with EC. In the case of the less stable surface (foam), the ratio in the evaluated effect reached the value of 70%, whereas muscle activity with EO on the unstable surface in 58% of cases corresponded with activity that was generated in the test with EC (Figure 3).

The interaction between the age of participants and the muscle activity during the test with EO and EC (Figure 4) turned out to be statistically important (*p* = 0.05). In the case of young people, there was a trend of increasing muscle activity in relation to surfaces of increasing instability. Similar EMG activities were observed in 85% of cases with EO and EC when standing at rest (on the ground), but the percentage of similar activities dropped to 72% when standing on the foam, and further dropped to 38% when standing on the seesaw.

A different situation was observed among the elderly participants. In comparing the tests with EO and EC, the ratio of muscle activity was lower than in the young participants and was similar in 92% of cases. The EMG activation ratio reached the value of 68% and 78% on the foam and seesaw, respectively.

The performance of normalization in the form of percentage values of relations between the examined factors (eyes, surface, age) does not directly show μV values in the activity of tested muscles and their differentiation in people of different ages and on different types of surfaces, therefore, the following results were presented in μV. 

### 3.1. Analysis of the EMG of Tibialis Anterior (TA) Muscle

In tests of both age groups, both with EO and EC, the unstable surfaces (foam and seesaw) forced higher EMG activation in relation to the stable surface (Figure 5). During the attempt with EO on the foam, the young participants generated higher EMG activation by 7 μV, whereas the older participants generated EMG activation that was 19 μV higher. The attempt on the seesaw resulted in a larger change in activity in relation to the stable surface, with the young participants registering 22 μV higher on the seesaw than the stable surface and the older participants registering as much as 49 μV higher. 

The young participants generated higher EMG activation in their attempts with EC on the unstable surfaces, with 27 μV on the foam and 115 μV on the seesaw. The EMG activation of the elderly participants was significantly smaller, with results higher by 37 μV on the foam and higher by 56 μV on the seesaw in relation to the stable surface. 

Among the young participants, only the test on the seesaw with EC showed a statistically significant difference in relation to the two other surfaces (i.e., the stable surface and foam). It was the surface on which TA activity was significantly different from the others (*p* ≤ 0.05). The result of the test on the seesaw with EO was significantly different from the test on the stable surface, whereas in the case of EC, both unstable surfaces were statistically different from the stable surface. 

### 3.2. Analysis of the EMG of Peroneus Longus (PL) Muscle

In the case of both age groups, average EMG values of the PL muscle showed a similar trend as in the case of the TA muscle, with notable differences in obtained values depending on the different factors (Figure 6). In tests of both age groups with EO and EC, the unstable surfaces (foam and seesaw) also forced higher EMG activation in relation to the stable surface. During the attempt with EO on the foam and seesaw, the young participants generated higher EMG activation by 7 μV in relation to the stable surface. The older participants generated 11 μV higher activity on the foam in comparison to the stable surface, whereas the results were higher by 19 μV on the seesaw in comparison to the stable surface. The young participants generated higher EMG activation in the attempts with EC on the unstable surfaces in relation to the stable surface, with 14 μV on the foam and 42 μV on the seesaw. EMG activation in the older participants increased by 12 μV on the foam in comparison to the stable surfaces, whereas their EMG activation on the seesaw increased by 11 μV in relation to the stable surface. 

In the case of the PL muscle, statistically significant differences were observed between the factors in the young and elderly participants, as was the case for the TA muscle.

### 3.3. Analysis of the EMG of Gastrocnemius Medialis (GM) Muscle

During the attempt with EO on the foam and seesaw, the elderly participants generated higher bioelectric activity by 19 μV in relation to the stable surface. The young participants generated activity higher by 14 μV on the foam in comparison to the stable surface, whereas their activity was higher by 12 μV on the seesaw in relation to the stable surface (Figure 7).

EMG activation in elderly participants on the foam increased by 27 μV in comparison to the stable surface, whereas on the seesaw it increased by 26 μV in relation to the stable surface. Young participants generated higher bioelectric muscle activity in the attempts with EC on the unstable surfaces in relation to the stable surface, with 30 μV on the foam and 46 μV on the seesaw.

Statistically significant differences (*p* ≤ 0.05) between the evaluated factors occurred only in the attempts with EC. Both of the unstable surfaces (foam, seesaw) were significantly different from the stable surface.

### 3.4. Analysis of EMG of Gastrocnemius Lateralis (GL) Muscle

In the case of both age groups, average EMG values of this muscle showed a similar trend, as in the case of the GM muscle, with notable differences in obtained values depending on the different factors (Figure 8). In the examined muscle, during the attempt with EO on the foam and seesaw, the younger participants generated higher EMG activation by 7 μV in relation to the stable surface. The elderly participants generated 6 μV higher activity on the foam in comparison to the stable surface, whereas it was barely 8 μV higher on the seesaw in relation to the stable surface. 

The young participants generated higher EMG activation in the attempts with EC on the unstable surfaces, with 6 μV on the foam and 14 μV on the seesaw. EMG activation in the elderly participants on the foam increased by 12 μV in comparison to the stable surface, whereas on the seesaw it increased by 11 μV in relation to the stable surface. 

With the young participants, only the test on the seesaw with EC showed a statistically significant difference in relation to attempts on the two other surfaces (i.e., the stable surface and foam). The result of the test on the seesaw with EO was significantly different from the test on the stable surface, whereas in the case of EC, both unstable surfaces were statistically different from the stable surface. 

The EMG activation of muscles stabilizing the ankle joint with EC was higher than in the case of EO, regardless of the surface type and age of participants. The greatest difference in EMG activation between EO and EC in the young participants was observed for attempts on the seesaw, whereas in the elderly participants the greatest difference was observed for attempts on the foam. The highest changes in EMG activation due to changes in the surface and visual perception were found in TA and PL muscles. Comparing both evaluated groups, the elderly participants had greater differences in EMG activity on the surfaces with EO than young people with EC. Attempts with EC caused more statistically significant changes in EMG activation. There were more of such changes in the elderly participants.

## 4. Discussion

To compensate for the deterioration of the posture control system, older adults show increased concurrent activity of the ankle muscles and stiffen the legs to counteract the increase of body rolling in difficult postures. The ankle stiffening strategy indicates a reduction in the degree of freedom accompanied by a higher risk for future falls in the older population. In this context, studies on older people revealed higher amounts of activity from tibial muscles in the front muscle activity [16]. Previous research suggests that ankle injuries can be caused by improper positioning of feet just before and during contact with the walking surface [17,18].

The comparative research conducted in this study on the groups of elderly and young participants showed that in the position of free standing on the hard stable surface, and on two types of unstable surfaces, the older participants generated higher EMG activation of the evaluated anterior group muscles (TA, PL). In the posterior muscles of the older participants, the tendency definitely changed in the case of the GL muscle, whereas the GM muscle showed very similar activity in both age groups. The results show that the posterior limit of stability is decreased within the elderly and, as a consequence, the center of gravity is moved away from it. This mechanism indicates that the ankle strategy is used by older people in order to keep a stable position [15]. During the tests on the unstable surfaces under the condition of limited eye perception (EC), the elderly participants generated significantly lower activity of the evaluated muscles as compared with the young participants. It was particularly visible in the attempt to keep a stable position on the seesaw, where the muscle activity significantly rose in the young participants, particularly on the medial limb side (TA, GM), whereas it stayed at the same level in the elderly participants. In addition, the increase in activity between the soft unstable surface (foam) and the hard unstable surface was of little importance. It must be mentioned that the elderly participants also had difficulties in detecting minor changes in the angle of the ankle joint, which exceeded 1°. This means that the neural system does not detect the changes in the ankle joint within this scope, which has the biggest influence on the location of the center of gravity in the standing position. This is, without any doubts, the main cause of the increased position instability in the late period of life [1,15]; moreover, the decreased muscle activation among the elderly can be a sign of processes of muscle excitement and contraction being slowed down [19]. In the present study, from the perspective of posturography, a change in the trajectory of the movement with characteristic oscillations of the projection of the center of gravity both towards sides as well as in the sagittal plane was observed. People develop a strategy of slowing their movements, which helps them to keep a static position [15], as a result of changes caused by body aging. As Horak highlights, the rehabilitation of older people, particularly with vestibular system disorders, should include training on unstable surfaces with exclusion of visual perception, but with head and trunk stabilization, in order to improve the effect of feedback [20].

Many elderly people have been shown to use a strategy that is less stable and has higher biomechanical costs [21,22,23,24]. The training of proprioception and the vestibular system, as well as strength and endurance training of lower limbs in addition to comprehensive recreational training, can improve the ability of elderly people to keep balance and lower the risk of falling [6,21,25].

It should also be highlighted that the changes connected with aging processes are high individualized, as people adopt different forms of strategies to protect against falling, however, the authors agree on the fact that the first collapse causes changes in body posture that are related to protection against the factor that could result in the next accident of this type [17,26].

Due to aging processes, elderly people often experience limitations affecting particular systems and structures responsible for keeping a vertical body position. There is a high degree of individuality both in terms of the everyday activities of elderly people as well as the choice of strategies used against loss of balance and protection from falling. The authors suggest though that the persons using the ankle strategy present a lower risk of falling [27,28,29]. Comparative studies between young and elderly people with the use of fMRI in response to vibrations showed a relation between brain regions associated with foot activity [30].

Comprehensive strength training of lower limb muscles, as well as proprioceptive training, can help the elderly improve their balance and can contribute to increased lower limb strength, even in the case of concurrent chronic diseases [25,31,32].

## 5. Conclusions

The results obtained in this work confirm the importance of proprioception training and muscle strength in the rehabilitation of older people as well as in terms of prophylactic exercise for preventing injury occurrence within the ankle. Ankle strategy training can be useful for the elderly population to improve the function of their lower limb muscles, thereby increasing their ability to keep their balance. Detecting functional deficits that predispose individuals to injury occurrence can allow for training to be tailored toward each individual, thus allowing them to make meaningful improvements to their current condition. Practical indications concerning performance and adjustment in regard to this type of prophylactic exercise require particular attention.

Due to the pilot nature of the study, the authors are aware that the tests were conducted on a small sample, which constitutes a limitation in the statistical analysis, as well as the participation of women only. Therefore, a study conducted on a larger group of participants of both sexes is suggested in order to provide more detailed information. In addition, further studies should include a larger number of muscles within the lower limbs and trunk.

## Figures and Tables

**Figure 1 ijerph-16-01544-f001:**
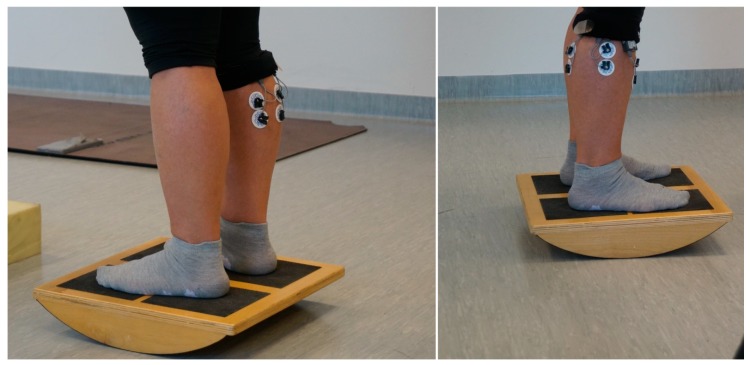
Illustration of EMG signal recording process.

**Figure 2 ijerph-16-01544-f002:**
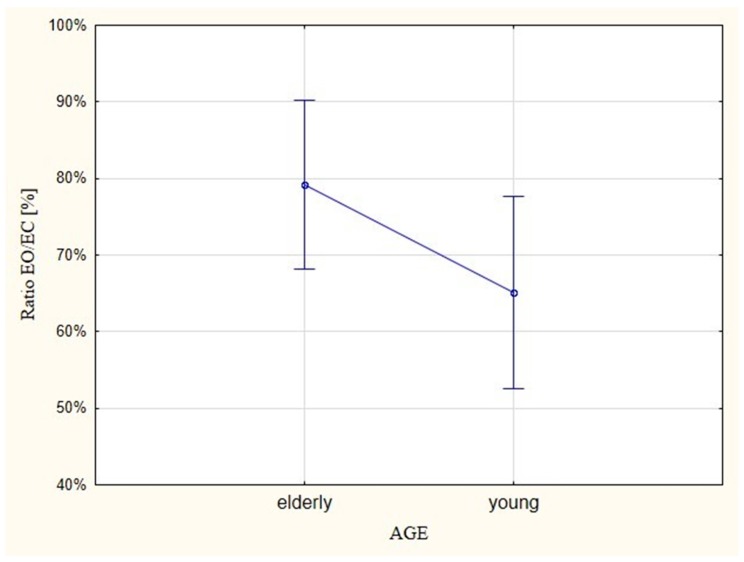
EMG ratios (ratios of synergistic and antagonistic muscles of the ankle joint) for eyes open (EO) versus eyes closed (EC) tests on young and elderly people.

**Figure 3 ijerph-16-01544-f003:**
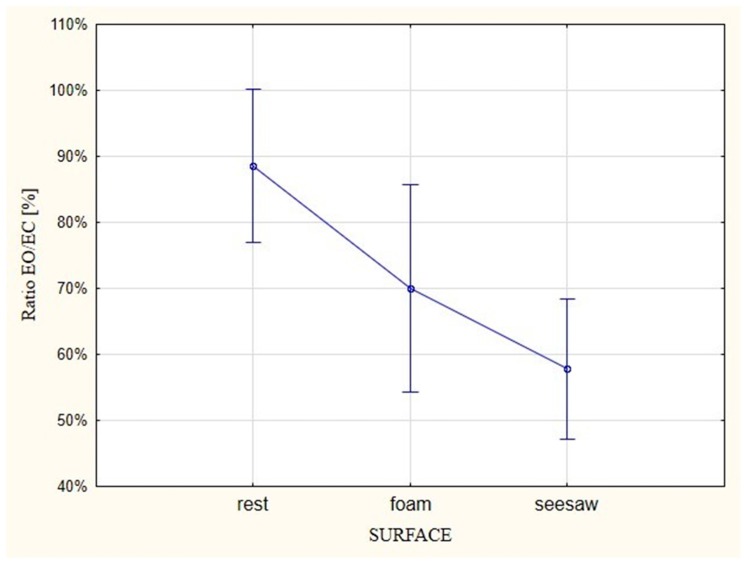
EMG ratios (ratios of synergistic and antagonistic muscles of the ankle joint) for tests performed with EO versus EC on different types of surfaces.

**Figure 4 ijerph-16-01544-f004:**
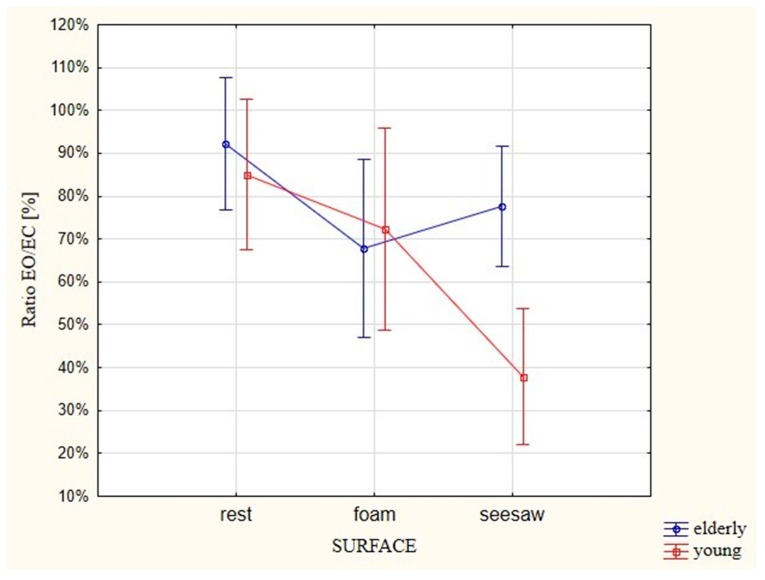
Comparison between groups of participants of different ages in terms of the EMG ratios (ratios of synergistic and antagonistic muscles of the ankle joint) for tests performed with EO versus EC on different types of surfaces.

**Figure 5 ijerph-16-01544-f005:**
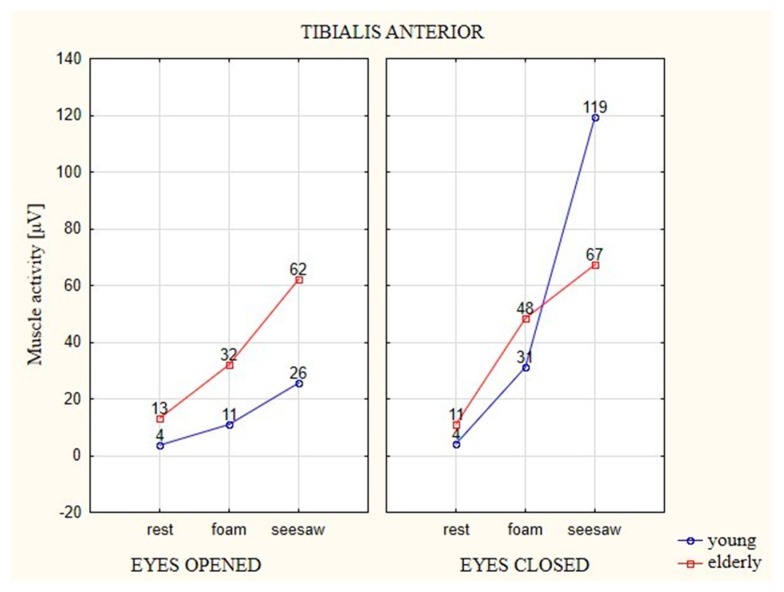
The average EMG values of the TA muscle in elderly and young participants on different types of surfaces and with EO (eyes 1) and EC (eyes 2).

**Figure 6 ijerph-16-01544-f006:**
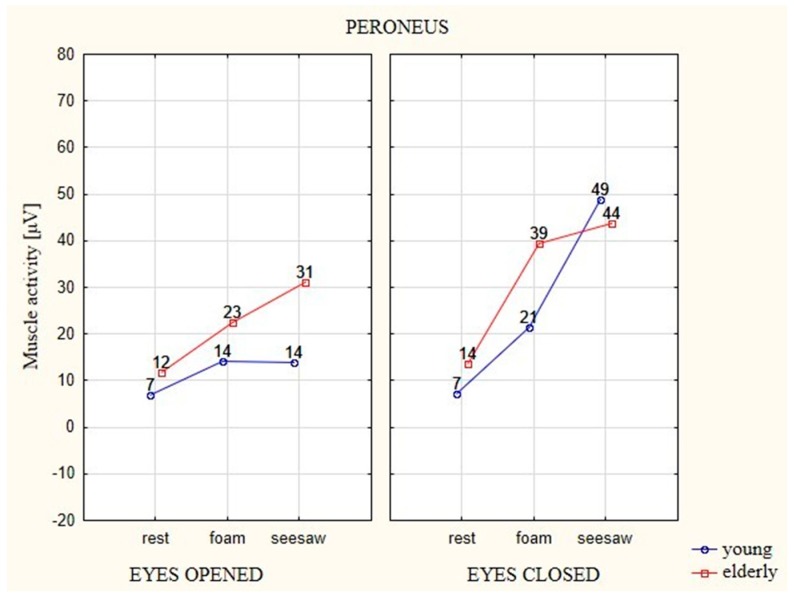
The average EMG values of the PL muscle in elderly and young participants on different types of surface and with EO (eyes 1) and EC (eyes 2).

**Figure 7 ijerph-16-01544-f007:**
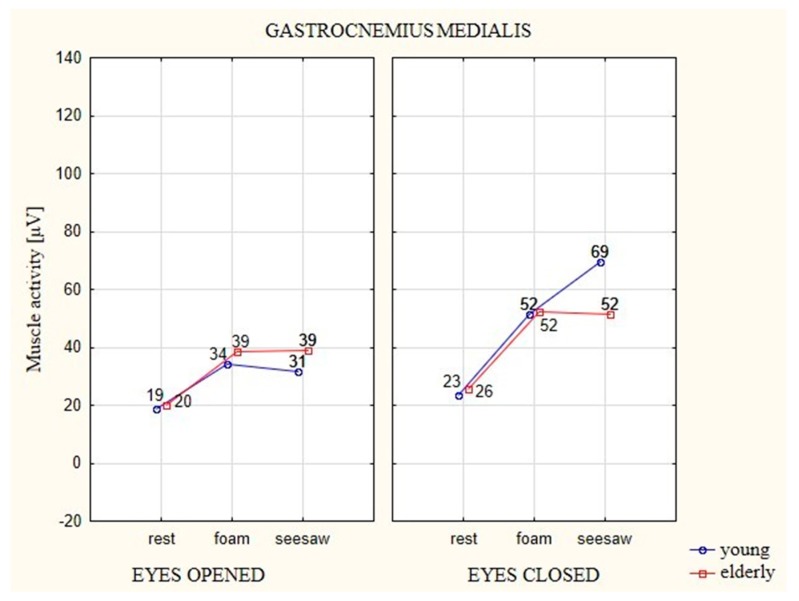
The average EMG values of the muscle GM in elderly and young participants on different types of surfaces and with EO (eyes 1) and EC (eyes 2).

**Figure 8 ijerph-16-01544-f008:**
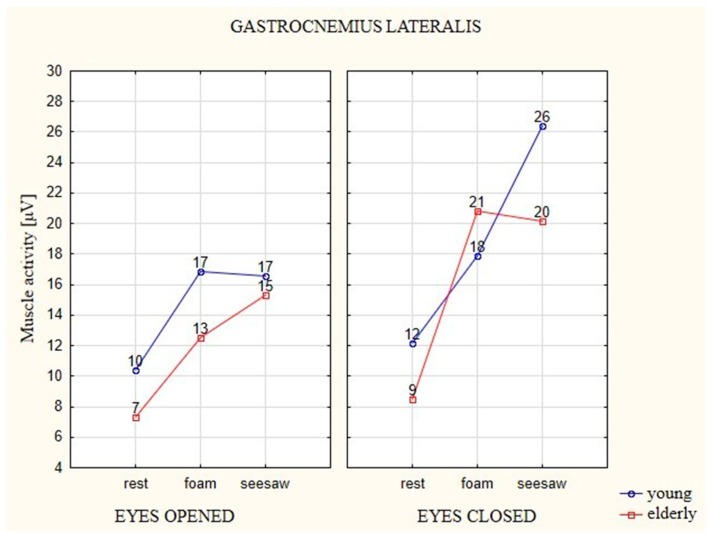
The average EMG values of the GL muscle in elderly and young participants on different types of surfaces and with EO (eyes 1) and EC (eyes 2).

**Table 1 ijerph-16-01544-t001:** Main effects and interaction of the analysis of variance.

Title	Age Effect		Surfaces Effect		Interaction: Surfaces × Age	
	F (1,14)	*p*	F (2,28)	*p*	F (2,28)	*p*
TA	3.155	0.097	0.880	0.426	0.268	0.767
PL	0.250	0.623	23.470	<0.001	7.970	0.002
GM	0.114	0.740	3.002	0.066	2.776	0.079
GL	0.184	0.675	3.903	0.032	4.905	0.015

TA—tibialis anterior, PL—peroneus longus, GM—gastrocnemius medialis, GL—gastrocnemius lateralis.

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
