# Peer review of "Analysis of Ankle sEMG on Both Stable and Unstable Surfaces for Elderly and Young Women—A Pilot Study"

_ijerph, 2019, doi:10.3390/ijerph16091544_

Round 1

Reviewer 1 Report

The main objective of this pilot study was to show the importance of EMG in the ankle joints on different surface forms with and without visual feedback. Participants were divided into two groups based on their age. Selected muscles activities were collected and compared in groups as well as conditions. The manuscript is well-written however I would encourage authors to incorporate following findings throughout the manuscript.

The main criticism is in the statistics. Since this study was considered as Pilot which was done with a limited number of subjects, p-values should be stated as needed. In addition, p should be adjusted since researchers compared a number of parameters and factors. I would encourage authors to justify the findings with adjusted p values respectively.

Line 2 Title, I recommend authors to mention “women” and “age group” in title.

Line 61. It is obvious that sEMG stands for surface EMG however I could not catch it in the manuscript before using it in abbreviation form.

Lines 61-63. Please indicate the refs for this statement.

Line 64-69. Authors mentioned that they accepted the hypothesis after stating the aim of this study. It is OK to design the experimental setup as they want but the story is kind of vague unless all the hypotheses and assumptions are well defined in order. I encourage authors to rephrase the last part of introduction accordingly,

Lines 71-72. Have you check of it is normal distribution since the number of subject is relatively small?

It is not stated but do you have subject demographics rather than gender and age, such as BMI? BMI is important in balance.

Lines 72-76. Did researchers test or verify participants’ declared health conditions or relied on what they said. It is important because such as hearing loss is one of the factors in elders falling. I assumed no one was deaf on the day of testing but wondering if researchers test participant’s eligibility.

Line 77. What do you mean by normalized attempts? Did subject attempt a few extra to get acclimated to the test?

Line 92. What type of electrode and brand did you use?

Line 100. What does “acc.” stand for?

Line 101. Is it one-standard deviation?

Line 120 Figure2. Please re-phrase the figure description. The figure indicates the EO/EC ratio in age groups. I recommend authors to indicate EO, EC elderly etc in the description. Also, the whiskers are one-standard deviation in addition to mean EO/EC ratio. I would say similar things for Figure3,4,5,6,7,8.

Author Response

We were happy to read the Reviewer's comments, which we appreciate. We have made our best efforts to make sure that the article and the changes introduced to it according to the Reviewer's comments as well as the answers to them will meet your expectations.

We are convinced that the content and technical aspects of the article have been improved.

Below, we refer to each of the Reviewer's comments and suggestions. The changes that have been introduced to the text are highlighted in red.

The main criticism is in the statistics. Since this study was considered as Pilot which was done with a limited number of subjects, p-values should be stated as needed. In addition, p should be adjusted since researchers compared a number of parameters and factors. I would encourage authors to justify the findings with adjusted p values respectively.

As suggested by the Reviewer, we completed the text and a description of the analyzes

Line 2 Title, I recommend authors to mention “women” and “age group” in title.

We agree with the reviewer's suggestions regarding the title of the article, which has been improved.

Line 61. It is obvious that sEMG stands for surface EMG however I could not catch it in the manuscript before using it in abbreviation form.

The text has been changed as suggested by the Reviewer.

Lines 61-63. Please indicate the refs for this statement.

We apologize for the oversight and the error in the test, the reference has been added.

Line 64-69. Authors mentioned that they accepted the hypothesis after stating the aim of this study. It is OK to design the experimental setup as they want but the story is kind of vague unless all the hypotheses and assumptions are well defined in order. I encourage authors to rephrase the last part of introduction accordingly,

Thank you for the suggestion, the text has been modified.

Lines 71-72. Have you check of it is normal distribution since the number of subject is relatively small?

To check the normal distribution, Shapiro-Wilk test was used.

It is not stated but do you have subject demographics rather than gender and age, such as BMI? BMI is important in balance.

We asked the subjects about the weight and height, according to the suggestion, the BMI was calculated and completed in the text.

Lines 72-76. Did researchers test or verify participants’ declared health conditions or relied on what they said. It is important because such as hearing loss is one of the factors in elders falling. I assumed no one was deaf on the day of testing but wondering if researchers test participant’s eligibility.

The group of young people were physical education students who declared high physical fitness, the elderly were chosen in accordance with the inclusion criteria given in the article. It consisted in conducting a short interview in accordance with the qualification questionnaire for the study carried out by a qualified physiotherapist.

Line 77. What do you mean by normalized attempts? Did subject attempt a few extra to get acclimated to the test?

Only in the case of the test on the seesaw, participants had the opportunity to repeat the test up to 3 times, in which case the best tasks were selected for the analysis, some subjects had trouble maintaining the seesaw in a neutral setting, ie when each edge didn't touch the ground.

Line 92. What type of electrode and brand did you use?

We used disposable electrodes (Bio-Lead-Lok B R-LFO-300) with a round pickup area, distance between electrodes 2.5 cm

Line 100. What does “acc.” stand for?

We apologize for the oversight and mistake in the test, they have been changed

Line 101. Is it one-standard deviation?

One-standard deviation

Line 120 Figure2. Please re-phrase the figure description. The figure indicates the EO/EC ratio in age groups. I recommend authors to indicate EO, EC elderly etc in the description. Also, the whiskers are one-standard deviation in addition to mean EO/EC ratio. I would say similar things for Figure3,4,5,6,7,8.

Thank you for the suggestion, we tried to explain it in the text

Reviewer 2 Report

In this manuscript the authors show that there are differences depending on the age in the bioelectric activity of some muscle of the ankle. This is important in order to apply measures in older people that could lead to a reduction of falls and other injuries.

Even though that is a pilot study I find that the sample size is very small. Moreover, the fact that all the participants were women, can also give a bias in the results. Why did the authors not consider the possibility of include men in the study?

As a pilot study, I suppose that the principal aim of the authors was to test the procedure in order to see problems, etc to be corrected for future studies with a larger size of persons. Did the authors found any particular thing that should be taken into account in this sense? If yes, they should include it in the discussion-conclusions.

Author Response

We were happy to read the Reviewer's comments, which we appreciate. We have made our best efforts to make sure that the article and the changes introduced to it according to the Reviewer's comments as well as the answers to them will meet your expectations.

We are convinced that the content and technical aspects of the article have been improved.

Below, we refer to each of the Reviewer's comments and suggestions. The changes that have been introduced to the text are highlighted in red.

Even though that is a pilot study I find that the sample size is very small. Moreover, the fact that all the participants were women, can also give a bias in the results. Why did the authors not consider the possibility of include men in the study?

We selected for research only women, because we wanted the group to be as homogeneous as possible. However, this was a pilot study, and taking into account the suggestions of the reviewer, future research will be conducted on a larger group of participants of both sexes, which will certainly provide more detailed information and potential changes within the given attempts.

As a pilot study, I suppose that the principal aim of the authors was to test the procedure in order to see problems, etc to be corrected for future studies with a larger size of persons. Did the authors found any particular thing that should be taken into account in this sense? If yes, they should include it in the discussion-conclusions.

After the pilot study, a few problems were found. Therefore we think, that a study conducted on the larger group of participants of both sexes should certainly provide more detailed information and potential changes within the given attempts. In addition, further studies should include the larger number of muscles within lower limbs and trunk.

Reviewer 3 Report

Summary:

Women of different age groups (mid 20s and mid 60s) performed balance testing on three surfaces: solid material, foam, and seesaw while their eyes were either open or closed. Muscle activities were recorded using surface electromyography.

Comments:

The authors have performed a preliminary study to examine the muscle activities of the anterior and posterior muscles controlling sagittal place ankle movements. There have been a number of studies looking at control of the ankle joint during different conditions, so it is unclear how this study adds to the current literature. The authors also need to assert the importance of this study and where it belongs in the literature.

The writing style is difficult to follow, particularly in the Introduction. A native English speaker should review the structure of the document to ensure proper wording and phrases.

The participant demographics are very limited (only age is provided). The authors have suggested that age is the primary difference between the participants’ results, but other factors, such as: total body mass, mass distribution, and height also provide potential variables influencing balance/stability of the body in these conditions.

There are no details regarding the parameters of the surface EMG signal (inter-electrode distance, CMRR, band-pass filter, skin preparation, etc.).

The commands EO and EC were used, during the balance trials, but it is unknown where the gaze was directed during the EO

A video camera was used during data collection, but it is not clear how this was used. Where markers placed on the participants and the movements tracked and analyzed in separate software? There is no mention of movement data in the Results section

The statistical analysis indicates a repeated measures ANOVA, but does not state the independent and dependent variables of concern. Since surface, EO/EC, and age were independent variables, why were no interaction effects tested? Level of significance?

In the Results section, the authors have compared EMG values between groups without establishing parameters of normalizing the EMG to compare between groups. This significantly undermines the interpretation of the EMG and invalidates the application of these data to the study in question.

The authors continuously refer to muscle activation as bioelectric activity. This is typically addressed as electromyography (EMG) in the literature and should be consistent with the literature.

In the Discussion/Conclusion section the authors allude to injury of the ankle influencing joint stability. This is difficult to assess in the study as participants were without lower extremity injury or neuromuscular diseases (p. 2, lines 72-75)

Specific comments:

p.1, lines 1-2: “the current population is aging rapidly” – what does this mean? The phrase is too colloquial

p.2, lines 84-85: “compensation in the knee and iliac” – what does this mean?

p.7, lines 213-214: what do you mean by “in the case of the elderly people, as many as three factors were different from the others”? The explanation for this is not clear

p.8, lines 231-232: ankle injury was not a factor for this study, so it is not clear why injury is mentioned in the opening paragraph of the Discussion section

p. 9, lines 272-273: “strategy present low risk of..” this sentence is incomplete

Author Response

We were happy to read the Reviewer's comments, which we appreciate. We have made our best efforts to make sure that the article and the changes introduced to it according to the Reviewer's comments as well as the answers to them will meet your expectations.

We are convinced that the content and technical aspects of the article have been improved.

Below, we refer to each of the Reviewer's comments and suggestions. The changes that have been introduced to the text are highlighted in red.

There are no details regarding the parameters of the surface EMG signal (inter-electrode distance, CMRR, band-pass filter, skin preparation, etc.).

Thank you for the suggestion. We included information about the research protocol in the article as suggested by the Reviewer

The commands EO and EC were used, during the balance trials, but it is unknown where the gaze was directed during the EO

The subjects were instructed to "look straight ahead”.

A video camera was used during data collection, but it is not clear how this was used. Where markers placed on the participants and the movements tracked and analyzed in separate software? There is no mention of movement data in the Results section

In the present study, we didn't used  motion capture systems were used, only a video camera  was used to evaluate posture and erroneous attempts during the examination

The statistical analysis indicates a repeated measures ANOVA, but does not state the independent and dependent variables of concern. Since surface, EO/EC, and age were independent variables, why were no interaction effects tested? Level of significance?

Thank you for the suggestion. We included information in the article and explained

In the Results section, the authors have compared EMG values between groups without establishing parameters of normalizing the EMG to compare between groups. This significantly undermines the interpretation of the EMG and invalidates the application of these data to the study in question.

Thank you for the suggestion, we tried to explain it in the text

The authors continuously refer to muscle activation as bioelectric activity. This is typically addressed as electromyography (EMG) in the literature and should be consistent with the literature.

According to the Reviewer suggestions, the text has been changed

In the Discussion/Conclusion section the authors allude to injury of the ankle influencing joint stability. This is difficult to assess in the study as participants were without lower extremity injury or neuromuscular diseases (p. 2, lines 72-75)

According to the Reviewer suggestions, the text has been changed

Specific comments:

p.1, lines 1-2: “the current population is aging rapidly” – what does this mean? The phrase is too colloquial

According to the Reviewer suggestions, the text has been changed

p.2, lines 84-85: “compensation in the knee and iliac” – what does this mean?

the text has been changed

p.7, lines 213-214: what do you mean by “in the case of the elderly people, as many as three factors were different from the others”? The explanation for this is not clear

This fragment is our oversight in terms of language and we apologize for this error, it has been corrected in the article.

p.8, lines 231-232: ankle injury was not a factor for this study, so it is not clear why injury is mentioned in the opening paragraph of the Discussion section

According to the Reviewer suggestions, the text has been changed

p. 9, lines 272-273: “strategy present low risk of..” this sentence is incomplete

We apologize for the oversight and the error in the test, the text  has been added.

Round 2

Reviewer 1 Report

I am satisfied with the response, don't have any further comments on this manuscript. Thank you.

Author Response

Thank you for your suggestions. We agree with the amendments and have included them in the manuscript. 

Reviewer 3 Report

p.2 line 95: "motion point" should be "motor point" or "innervation point"

p.3, line 11: Statistica 13.0 is referenced, but line 105 indicates Statistica 12. Which version was used?

Table 1: the abbreviations need to be defined in a footnote - are these muscles groups assessed with the sEMG?

Figures 2, 3, and 4: which muscle is shown for this illustration? did the authors composite the EMG?

Author Response

We agree with the reviewer's suggestions and have amended the text (amendments
marked in blue)

kind regards
